# Evaluation of the Pre-Planned and Non-Planed Agility Performance: Comparison between Individual and Team Sports

**DOI:** 10.3390/ijerph17030975

**Published:** 2020-02-04

**Authors:** Krzysztof Mackala, Janez Vodičar, Milan Žvan, Jožef Križaj, Jacek Stodolka, Samo Rauter, Jožef Šimenko, Milan Čoh

**Affiliations:** 1Department of Track and Field, University School of Physical Education, Wroclaw, Ul. Paderewskiego 35, 51-612 Wrocław, Poland; jacek.stodolka@awf.wroc.pl; 2Faculty of Sport, University of Ljubljana, Gortanova ul. 22, 1000 Ljubljana, Slovenia; Janez.Vodicar@fsp.uni-lj.si (J.V.); milan.zvan@fsp.uni-lj.si (M.Ž.); jozef.krizaj@fsp.uni-lj.si (J.K.); Samo.Rauter@fsp.uni-lj.si (S.R.); jozef.simenko@fsp.uni-lj.si (J.Š.); Milan.Coh@fsp.uni-lj.si (M.Č.); 3School of Life and Medical Sciences, University of Hertfordshire, Hatfield AL10 9EU, UK

**Keywords:** agility, reaction time, CODS, testing, athletic performance, cognition

## Abstract

This study assessed differences in agility performance between athletes of team and individual sports by assessing change-of-direction speed (CODS) as pre-planned agility and reactive agility (RA) as non-planed in different spatial configurations. The study involved 36 individual (sprint, hurdles, jumping, tennis, and judo) and 34 team (soccer, basketball, and handball) athletes. CODS and RA were measured with a light-based reactive training system in a frontal (FR), universal (UN), semicircular (SC), and lateral (LA) design. Lower limb power and sprint performance were also measured in a 10 m single leg jump test and 15 m sprint. Individual athletes showed significantly better performance in three of the eight agility tests: LA-RA, UN-RA, and SC-CODS (*p* < 0.008, *p* < 0.036, and *p* < 0.027, respectively) and were found to present stronger correlations (*p* < 0.01) between jump test performance and the CODS condition. Team athletes showed stronger associations between sprint performance and the CODS condition. In the RA condition both jump and sprint performance showed stronger correlations in the group of individual athletes. Agility performance as measured by CODS and RA should improve with enhanced of motor proficiency. Finally, the tests applied in this experiment seem to be multidimensional, but require spatio-temporal adjustment for their implementation, so that they meet the requirements of the particular sport.

## 1. Introduction

The particular demands of a sport are reflected in the physical, technical, and tactical abilities of athletes [1]. These abilities are abetted by a particular anthropometric or physiologic profile encompassing a wide variety of variables from body height, body type, and muscle fiber composition to metabolic capacity, muscle contractile properties, or motor proficiency. In addition to these characteristics, there are also a number of psychological factors that dictate performance such as athlete motivation, cognitive skills, and emotional intelligence [2]. All of the above factors influence the rate at which an athlete can achieve mastery and can be treated as determinants of competitive success. 

A sport can generally be distinguished as an individual or team sport, in which performance is dictated exclusively by the effort of the individual compared with the collective interaction of multiple teammates (or at least two individuals). Other more complex categorizations are based on the division of sports by their motor and technical characteristics or physiological demands. While individual sports such as singles tennis or judo are recognized as individual sports compared with soccer or basketball, all of these disciplines are intermittent in activity whereby athletes are required to frequently transition between brief bouts of high-intensity activity and longer periods of low-intensity activity [3,4]. In contrast, some of the sports that comprise individual track and field events such as sprinting, hurdling, or jumping all rely on speed and power, whereas others are based entirely on endurance (distance running) or strength (throwing events). On this basis, several sports categories have been identified: target games, net/wall games, striking/fielding games, and territory games [5]. One uniting factor is that almost all individual and team sports require a great deal of technical proficiency in changing direction, speed, and position. Known as agility, many of these movements comprise a “hinged moment” condition involving both linear and angular momentum [6].

The criteria that guided the division of individual disciplines into team sports and individual sports are based on the individual entity (player) who implements the goals set by individual sport performance. He is a single perpetrator of the action (individual sport) or cooperates with many entities (players) and pursues their common goal of particular sport performance (team sports). According to Kłyszejko [7], the analysis of factors influencing the course of sports performance and assessment of movements structure, that actually apply and play a specific role in this performance, allows the classification of sport’s disciplines according to their movement separateness. There is no strong articulation that the course of the sports performance, as well as the effects of the athlete’s actions, depend on the level of motor abilities. In this context, it seems obvious that the sports result (performance) is also influenced by motor skills. Based on the thoughts of Kłyszejko [7], Naglak [8,9] took into account the subjectivity and manner of achieving the objectives of sports performance, and on this basis distinguished the sport disciplines on individual, group, and team disciplines.

In individual sport disciplines (all types of races, wrestling, fencing, judo, gymnastics, sprints, various types of jumps, tennis) only single athletes take part. He/she relies on their own technical skills, and the performance requires them to be physically active and ready to take action to achieve the performance goal-victory. When two or more athletes compete together with other athletes, we are talking about the group type of sport disciplines. This is characterized by assigning specific positions of each athlete in the group or specific responsible roles which they need to perform. In addition to psycho-physical requirements, sports performance requires from each athletes the ability to predict the behavior of a partner and competitors. It occurs during rowing and canoeing races, relay races, and when playing in pairs e.g., in badminton or tennis. 

In team disciplines, a certain number of athletes must participate in sports performance. Each member of the team carries out the assigned tasks and cooperates with partners to achieve the set goals. The team enhances its effectiveness through appropriate organization of conducting sports performance (games), taking into account the goals of particular sports, and players’ dispositions, opponents’ dispositions, thus using the capabilities of individual athletes to achieve the best synchronization of sport actions. There is no perfect division of sport into disciplines and there will always be some exclusion or departure from the rule, but it does not change the basic fact of some division and the correctness of its methodological use.

Generally, the definition of agility is simple, and at the same time its context is very complex, which means that it can be widely used in practice. This manifests itself in both the selection of agility exercises and assessment tests. In all of the above sports (except of track and field), agility plays a major role in both motor preparation and competition. Its manifestation may vary, but the ability to start, stop, change the direction of displacement, and restart requires the implementation of the same movement structures.

Agility is a significant component of athletic performance as it encompasses the ability to perform whole-body and local movements rapidly, effectively, and efficiently [10,11,12,13,14]. While the classic definition of agility is understood as the ability to change the direction or speed of a movement both quickly and precisely, more recent definitions have integrated a cognitive component. This encompasses factors such as anticipation, visual and perceptual recognition, reaction speed, and attention skills [14,15]. Both agility definitions are correct and determine their significance and usefulness in selected sports. Their application, however, depends on what agility component we measure and evaluate. Consequently, movement scenarios that involve a reaction to an unplanned or random stimulus are known as reactive agility (RA), whereas pre-planned movement situations with no stimulus are determined more by change-of-direction speed (CODS) without any reactive or cognitive component [14,15]. In sports situations, RA is more common than pre-planned CODS since sport-specific situations are frequently unpredictable and determined by the movement or actions of an opponent or object (e.g., a ball) and therefore require a cognitive response [16]. Several authors have reported that reactive agility is a significant criterion of performance in football, handball, basketball, volleyball, hockey, rugby, and other team sports [12,15].

An important element in assessing the level of agility are agility tests. They are inseparably associated with the ability to accelerate, decelerate (slow down) or almost completely stop, and restart, or change the direction of running at different angles. Depending on the spatio-temporal configuration of the test, the time it takes to overcome depends on many factors, but mainly on the level of basic motor skills and body composition including body mass, body height, or length of the lower limbs [17,18,19,20,21]. Additionally, considering the dynamics of each sport, it is likely that athletes of individual and team sports would show differences in the non-planned reactive component of agility (RA) compared against planned and anticipated changes in either direction or speed (CODS). Surprisingly, no studies to date have compared agility performance between athletes of team and individual sports. Considering the importance of CODS and RA for successful performance, the aim of this study was to assess differences in agility performance between athletes of team and individual sports by assessing change-of-direction speed (CODS) and reactive agility (RA) in different spatial configurations. 

Neither individual nor team athletes encounter a light-based stimulus in their respective sport although track athletes, namely sprinters and hurdlers, need to instantaneously respond to the auditory signal of a starting pistol [22,23,24]. However, this is a simple stimulus compared with the more complex visual, auditory (verbal communication), and kinesthetic stimuli experienced by athletes when confronting an opponent [25,26,27,28,29]. Furthermore, the group of individual athletes also included tennis players and judokas who, like team athletes, react to the external stimuli generated by an opponent. As the present study involved a stimulus unknown to both groups of athletes, it was inferred that the type of external stimuli would be a controlled factor [15,30]. Therefore, it can be hypothesized that team athletes would show enhanced performance in agility tests with a RA component due to the greater number of random stimuli experienced by this group. 

## 2. Methods

### 2.1. Subjects

The study involved 36 athletes of individual sports and 34 athletes of team sports. The distribution of sex across the groups of particular sports was not equal. In game sports there were 20 males and 14 females (soccer: *n* = 7/9, basketball: *n* = 9/3, and handball: *n* = 4/2, respectively). In turn, the individual athletes were divided into 24 males and 12 females (track-sprint, hurdles, jumping: *n* = 15/9, tennis: *n* = 5/2, and judo: *n* = 4/1, respectively). Individual athlete mean age was 20.58 ± 1.44 years, body height was 174.08 ± 8.41 cm, body mass was 68.77 ± 11.34 kg, and lower limb length was 97.89 ± 5.87 cm. For team athletes mean age was 21.31 ± 1.72 years, body height was 179.21 ± 10.85 cm, body mass was 78.13 ± 12.39 kg, and lower limb length was 101.23 ± 7.02 cm. Participants were recruited from local university teams or local sport clubs. All had a minimum of 4 years competitive experience, trained regularly (3 to 4 sessions per week), and were free of injury or illness. All had experience in plyometrics (unilateral or bilateral), linear speed, and agility training. As the participants competed in different disciplines, they were at different phases in the training macrocycle. The team athletes were in the pre-competitive and competitive season, track and judo athletes were in the preparatory phase, and the tennis players in the pre-competition phase. In addition, while both males and females were recruited and sex differences in CODS and RA had been previously reported [14], this variable was not analyzed due to the unequal distribution of sex (41 male and 29 female). The participants were informed about the purpose and procedures of the study and written consent was obtained. Participation was voluntary and could be terminated at any time. The study design was approved by the Human Ethics Committee of the University of Ljubljana (Code: 14_2019-1433).

### 2.2. Procedures

Agility in the CODS and RA condition was measured using the FitLight Trainer (Sport Corp., Ontario, Canada). This is a reaction training system composed of a wireless controller and several lights placed on 40 cm cones. The lights can be arranged in various spatial configurations and coded for specific light activation sequences. The light can then be turned off by a proximity sensor located on the light. Testing began from a centrally placed light. The participant would wave their hand over the central light to activate one of the lights according to the sequence required by the CODS or RA condition. The participant would sprint to the light and deactivate it, then return and deactivate the central light and then sprint to the next activated light. The test ended when the participant deactivated all the lights and returned to the central light. Performance was measured by the time to completion. 

In order to provide a comparative task in the CODS and RA condition across a variety of movement scenarios, a frontal (FR), universal (UN), semi-circular (SC), and lateral (LA) spatial configuration was applied in accordance with previous research [14]. The arrangement and distances between the lights are presented in Figure 1. Due to number of participants and number of trials, testing occurred over four sessions in a period of 1 month where frontal change-of-direction speed (FR-CODS) and frontal reactive agility (FR-RA) were assessed in the first session, universal change-of-direction speed (UN-CODS) and universal reactive agility (UN-RA) in the second session, semicircular change-of-direction speed (SC-CODS) and semicircular reactive agility (SC-RA) in the third session, and lateral change-of direction speed (LA-CODS) and lateral reactive agility (LA-RA) in the fourth session. An additional testing session evaluated the participants for sprint and jump performance.

#### 2.2.1. Agility Testing 

A familiarization session was administered 1 week before the first session in order to acquaint the participants with the procedures and FitLight Trainer system. In each session the CODS condition was performed first and then the RA condition second after approximately 60 min of rest. The procedure for the pre-planned CODS condition involved activating the lights in sequence with the participants given advanced notice on the order (FR: 1-2-3-4-5-6, UN: 1-2-3-4-5-6, SC: 1-2-3-4-5, LA: 1-2-3-4). Two trials were performed (separated by 90 s of rest) and the best time to completion was selected. The same configurations were used in the unplanned RA condition although the order of light activation was non-sequential and unknown to the participants (FA: 2- 6-4-3-6-1, UN: 6-3-2-4-1-5, SC: 4-2-3-1-5, LA: 3-1-4-2). Similar in the CODS condition, two trials were completed and the best time to completion was recorded. A standardized warm-up was performed before each trial.

#### 2.2.2. Sprint and Explosive Power Testing 

A 15 m straight-line sprint was used to measure sprint performance from a standing start (S15m) and a flying start (F15m). An indoor track was equipped with timing gates (Brower Timing System) at the start (0 m) and finish (15 m) lines. For the flying start a 20 m approach was used to ensure attainment of maximal speed prior to entering the 15 m stretch. The participants completed two trials for S15m and two trials for F15m (with a 2 min rest between each trial) and the shortest time was selected for data analysis. Prior to sprinting the participants performed a short warm-up of light jogging, stretching, skipping drills, and several 30–50 m sprints. Upon completing the sprint component, the participants rested for 5 min and then completed the 10 m single leg jump test (SLJ10m) on the same indoor track to assess lower limb explosive power. Starting from a stationary position, the participant stood on one leg in front of the timing gate and was to jump as quickly as possible across the 10 m distance. Participants performed two trials for each leg with each trial separated by 3 min of rest. The best time to completion was selected. 

### 2.3. Statistical Analysis

Data analysis was performed with SPSS for Windows 15.0 (IBM, Armonk, NY, USA). Descriptive statistics (mean ± SD) were calculated for all dependent variables. Between-group comparisons were made with Student’s *t*-test for independent samples. Because of the normal distribution, Fisher’s Least Significant Difference test was applied post hoc to calculate the pairwise differences when significant F ratios were obtained. Pearson’s product-moment correlation analysis was used to assess the linear relationship between the variables. Effect sizes were evaluated by calculating Cohen’s d with 95% confidence intervals. Cohen suggested that *d* = 0.2 be considered a ‘small’ effect size, 0.5 represents a ‘medium’ effect size, and 0.8 a ‘large’ effect size. Statistical power was at the 0.90 level and significance was accepted at alpha = 0.05. The test–retest reliability of the agility and motor tests was evaluated by intraclass correlation coefficients (ICC). All four configurations in the CODS condition were found to have strong reliability with the highest values obtained for FR-CODS and LA-CODS (*r* = 0.88 and *r* = 0.91, respectively). The RA condition showed less reliability (*r* = 0.81–0.85) with UN-RA showing the strongest reliability (*r* = 0.85). ICC for the S15m and F15m tests were *r* = 0.86 to *r* = 0.90, respectively, whereas the ICC for the SLJ10m was *r* = 0.92. 

## 3. Results

No between-group differences were observed for age (Table 1). A statistically significant difference was observed for body height (2.9% differences), and body mass and lower limb length differed by 13.6% and 3.4%, respectively. Significant differences between the groups were found in three of the eight agility tests (UN-RA, SC-CODS, and LA-RA), with individual athletes outperforming team players (Table 2). The greatest absolute difference was in LA-RA by 1.07 s (7.8%) and then in UN-RA by 1.04 s (6.5%) and SC-CODS by 0.85 s (4.9%). Individual athletes also showed enhanced sprint (S15m) and jump performance (SLJ10m), with the greatest between-group difference in right and leg SLJ10m by 10.1% and 9.1%, respectively (Table 3).

Significant differences were observed in both groups between the CODS and RA condition for each of the spatial configurations (Table 4), with the greatest difference in the UN configuration (by 31.1% for individual athletes and 20.9% for team athletes), FR configuration (by 15.3% for team athletes), and LA configuration (by 14.9% for individual athletes). The smallest difference between CODS and RA performance was in the LA configuration (by 10.3% for team athletes and 12.7% for individual athletes).

Correlation analysis revealed strong positive correlations between CODS and RA performance in the majority of the spatial configurations (Table 5). The strongest correlations were observed in individual athletes for the LA and FR configurations at *r =* 0.89 and *r* = 0.80, respectively. Significant correlations were also observed between CODS and RA performance, sprint and jump performance, and a variety of the anthropometric characteristics (Table 6). Body mass was significantly correlated with CODS and RA performance in both groups. Strong negative associations were found between body mass and SC-CODS in team athletes (*p* = 0.40), and FR-CODS and LA-CODS in individual athletes. Stronger correlations for both CODS and RA were observed with body height, with the strongest correlation for UN-CODS and LA-CODS (*r* = 0.40, *r* = 0.55, respectively) in team athletes. The jump test measuring lower limb explosive power was also strongly correlated with CODS performance for UN-CODS and LA-CODS, particularly in team athletes for both limbs (left and right). In turn, individual athletes showed a correlation only with UN-RA. A similar trend was also observed in both sprint tests (S15m and F15m), where stronger correlations were found with 15 m sprint form flying start with CODS performance among the team athletes, whereas stronger correlations were found with RA performance among the individual athletes for UN-RA and LA-RA (*r* = 0.46 and *r* = 0.50 respectively).

Correlations between the spatial configurations for each of the agility conditions revealed few strong associations. In the CODS condition, the strongest correlations were between FR-CODS, SC-CODS, and LA-CODS in team athletes (*r* = 0.62, *r* = 0.45, and *r* = 0.60, respectively) and between FR-CODS and UN-RA in individual athletes (*r* = 0.48). In the RA condition, correlations were found between LA-RA and UN-RA and SC-RA in group of individual athletes.

## 4. Discussion

Individual athletes showed better agility performance than team athletes as evidenced by the significant differences observed in three agility tests (LA-RA, UN-RA, and SC-CODS). This result was particularly surprising in regard to the RA condition and implicates that athletes of individual sports may show an improved enhanced reactionary response, cognitive processing, or better conditioning which translate to improved acceleration, braking, dynamic balance, and change of direction or speed ability. 

The obtained data is the result of the application of group analysis (comparison between two groups of sport disciplines). This is a fact. However, explaining why one group of athletes achieved better results than the other in individual tests requires a slightly different approach. To answer the question why this phenomenon took place and what could cause it, one should delve deeper into the analysis of individual motor structures that meet the definition of agility. This means that differences in results between groups must be analyzed based on internal analysis within the group, which take into account a comparative group analysis. However it should have more general character. Both approaches are not mutually exclusive, they complement each other, even though we can get the impression that it is methodologically incorrect

Therefore, various sports disciplines, can be assessed in terms of agility level with the same test, despite the fact that there is a variety of movement structures defining agility as a motor ability. The ability to start, stop, change the direction of displacement, and restart requires the implementation of the same movement structures. They are different only in time-space configuration. For example, judoka needs one, or maybe two small steps, taken quickly and dynamically to attack the opponent and reach the throw position; conversely a tennis player needs 4–5 steps to reach the coming ball [31] and a soccer player needs a dozen or so [32]. Therefore, everyone needs a start that implements a special movement structure that must be learned. When we need to stop to do another movement—for example judoka jump away from the opponent, the tennis player returns to hit the ball again [31], and the basketball player makes a throw [17]—it requires the stopping ability and its optimal performance in terms of technique and efficiency. It is similar with changing the direction where, judoka will do it in one, or two small steps with a choice of left or right turn, tennis in one direction moving toward incoming ball, and the basketball player due to the action of the opponent, partners or moving ball multi-direction. Everyone must complete a change-of-direction movement and restart maneuver. These elements should be learned, practiced, and checked by appropriate test for correctness and effectiveness of performance.

In retrospect, the measurement of the time from light activation and initial movement to light deactivation would have allowed us to discriminate the groups by reaction time. Unfortunately, this reaction time was not measured and future research ought to include this valuable measure as well as determine if other forms of stimuli may influence response time and execution. However, if we can assume that both groups exhibited a similar response time, it is possible that the difference in performance between both groups in LA and UN configurations can be explained by the technical difficulty of the test regardless of the RA or CODS condition [15,31]. When executing the LA configuration, the participant has to complete a series of side shuffles. This movement structure may be surmised as less natural for track and field athletes than for handball players and basketball players, whom frequently perform this type of multi-directional activity [32]. However, individual athletes such as tennis players and judokas not only frequently perform lateral shuffles, but need to perform this movement very frequently which may have influenced this group’s performance in the LA configuration [33].

Regarding the UN configuration, this spatial scenario can be recognized as the most complex of the four as it requires significant visual perception, concentration, and movement dynamics [14,16,34]. While team athletes may have greater experience with performing movements associated with the UN structure, it is possible that individual athletes showed better cognitive processing during task execution [23,35]. Previous studies have highlighted this difference where peripheral perception and other cognitive components are more critical for effective performance than in team sports in which the final results rely on teamwork [2,35]. Therefore, it seems that the spatio-temporal variability in implementing agility in a particular discipline had a significant impact on obtained results in applied agility tests. We must remember that these tests had the same configuration but were performed as non-planed implementing RA and planned implementing CODS. In this case, with the time–space restriction of movement execution (1–2 steps, 3–5 steps, and so on), it should be assumed that a small group of (4) judokas and slightly larger tennis players (6) did not show much influence on the results obtained. It should be assumed that track and field athletes decided about the results, despite the fact that they do not use agility in competition and training. To a large extent, the tests were based on a straight run (different lengths of sections) with the initiation of a quick start. In this respect track athletes have much more experience and above all motor potential to achieve better times. However, each run ended with braking and change of direction and a move either sideways, diagonally, or backwards. In this element, athletes show little skill, but tennis players deal with this element very well. This can also support better performance by individual sports groups.

Another explanation for the difference in agility performance between the two groups may lie with the motor potential of the athletes. In the assessments of sprint and jump performance, individual athletes outperformed team athletes in the RLJ10m and LLJ10m by 10.1% and 9.1%, respectively, and had better times in the S15m. The majority of the participants who comprised the group of individual athletes were sprinters, hurdlers, and jumpers, for whom a fundamental aspect of training is the improvement of linear sprint performance and lower limb power and strength [35,36,37]. As a result, the enhanced level of lower limb explosive power and sprint acceleration may have been more important than various cognitive aspects or the spatial configuration of the agility tests. This may contradict the findings of Holmberg [11] who concluded that straight-line sprinting does not translate to enhanced agility performance and instead supports the position of Popowczak [38] in that linear running speed and jumping ability are important determinants of CODS.

Correlation analysis revealed a dependency between performance in the single leg jump and agility tests, confirming the conclusions of previous studies in which this association is due to the similar biomechanical foundation of the tested movement structures—rapid change in eccentric/concentric contractions and short response times [14,39]. These associations differed between groups, in which individual athletes showed stronger correlations between jump performance and RA, whereas team athletes showed stronger correlations between sprint performance and CODS. Both jump and sprint performance were more strongly correlated with the CODS tests in the group of team athletes. 

We also cannot forget about the connections between body structure and performance of agility [17,18,19,20,21]. In our research, only a few tests carried out in two CODS and RA configurations showed significant relationships with body weight, body height, and leg length (Table 6). Statistically significant relationships were confirmed in both groups and concerned body length factors in the area of planned change of direction (CODS), respectively: UN-CODS, SC-CODS and LA-CODS, with an indication of players from team games. These values did not exceed *r* = 0.60. In turn, body weight showed two negative relationships related to RA and CODS tests among individual sports. There is no reference in the literature to these data due to the lack of repeated tests in this spatio-temporal configuration. However, the results suggest that the more complex the movement structure during agility testing, the stronger the relationship between motor proficiency and RA and CODS performance is.

Finally, after a comprehensive analysis of the collected material, it can be seen that, there are significant differences in the times of particular agility tests between both groups of disciplines. This applies to both CODS-related tests and RA tests. There are also significant differences between the individual sport athletes and team games players when we analyze the relationships between special motor ability tests and the results of agility tests. Individual athletes showed stronger correlations between jump performance and the RA condition, whereas team players showed stronger correlations between sprint performance and CODS. In the RA condition, stronger correlations with both jump and sprint performance were found in individual athletes. This indicates that the results in both tests for the assessment of agility level (times), as well as the improvement in the performance of movement structures determining the level of agility, depend on the general level of other motor skills, mainly power and speed.

The present results need to be interpreted with caution as the study featured certain limitations. First, as previously acknowledged, was the lack of measuring reaction time from the moment of light activation and movement initiation to light deactivation as it would have allowed us to examine agility performance in more detail. Second, while the assessment of athletes according to the generalized category of individual and team sports appears to be valid, future research should refine or even further differentiate the groups to different criteria as the groups included athletes for whom agility is a far more important quality than in others. Therefore, when it comes to team games, this division is quite often used, despite the differences in terms of play game, technique and motor ability requirements. Individual sports show too many differences that limit their integration into one common dimension. In future scientific research, only two sports should be analyzed and compared, e.g., one team game and one of individual sports, or a comparison of each game from team sports or comparison of track and field to another sport. Third, because of the high number of participants from different sports and number of trials, testing occurred over four sessions in a period of 1 month. It can may create a problem because the athletes during this period was exposed to many training factors that may impact their physical and psychological status. A more in-depth comparison of RA and CODS performance between athletes should involve larger groups of homogenous athletes in terms of sport, performance level, experience, and sex. Another division of sport may be a limiting factor. It also seems reasonable to analyze and demonstrate differences athletes who train for sport that include change of direction speed and reactive agility ant those that do not.

## 5. Conclusions

There are differences in the results achieved between the group of sport discipline for individual and team in particular agility tests. This indicates that each group achieved better results in specific tests This applies to both CODS-related tests and RA tests. It should be assumed that one of the elements affecting this phenomenon is the spatial-temporal configuration of agility testing, which in most cases also differs from the spatial-temporal movement structures determining the level of agility in a given sport discipline, except for track and field. 

The second important factor is that, the differences in the results between the two groups of disciplines may result from the level of basic motor skills such as strength, power of the lower limbs, or speed, which undoubtedly affects the level of tasks carried out within agility. Here we should distinguish track and field athletes who, thanks to their high level of motor ability, have contributed to achieving better results in selected tests. This suggest that regardless of the practiced sport or discipline, agility performance as measured by CODS and RA could be better enhanced by improving motor proficiency.

The third conclusion and probably the most important is that the tests applied in this experiment seem to be multidimensional, but require spatio-temporal adjustment for their implementation, so that they meet the requirements of the particular sport.

## Figures and Tables

**Figure 1 ijerph-17-00975-f001:**
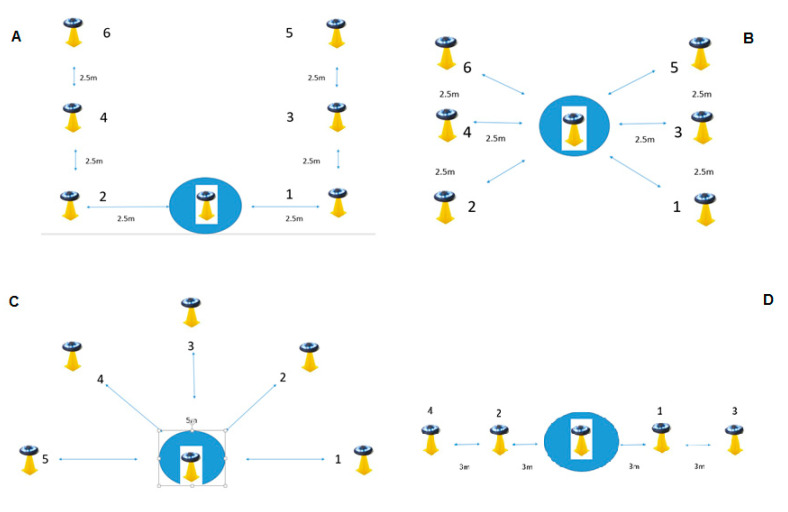
Arrangement of FitLight Trainer lights in the frontal (**A**), universal (**B**), semi-circular (**C**), and lateral (**D**) spatial configurations.

**Table 1 ijerph-17-00975-t001:** Age and anthropometric characteristics and between-group comparisons.

Variables	Group	*N*	Mean	SD	95% Confidence Interval of the Mean		*d*	*F*	*p*
					Lower	Upper			
Age [years]	Individ.	36	20.58	1.44	20.10	21.07	0.47	3.83	0.055
	Team	34	21.32	1.72	20.72	21.92			
Body height [cm]	Individ.	36	174.08	8.41	171.24	176.93	0.54	4.90	0.030
	Team	34	179.21	10.85	175.42	182.99			
Body mass [kg]	Individ.	36	68.77	11.34	64. 40	72.61	0.80	10.87	0.002
	Team	34	78.13	12.39	73.80	82.45			
Lower limb length [cm]	Individ.	36	97.89	5.87	95.91	99.88	0.53	4.68	0.034
	Team	34	101.23	7.02	98.78	103.68			

**Table 2 ijerph-17-00975-t002:** Change-of-direction speed (CODS) and reactive agility (RA) performance in the frontal (FR), universal (UN), semi-circular (SC), and lateral (LA) configurations and between-group comparisons.

Variables	Group	*N*	Mean	SD	95% Confidence Interval of the Mean		*F*	*p*	*d*
					Lower	Upper			
FR-CODS [s]	Individ.	36	16.68	1.77	16.08	17.28	1.51	0.223	0.30
	Team	32	16.16	1.75	15.53	16.78			
FR-RA [s]	Individ.	35	19.08	1.87	18.44	19.73	0.97	0.328	0.25
	Team	31	18.63	1.87	17.94	19.32			
UN-CODS [s]	Individ.	31	13.01	1.38	12.51	13.52	0.37	0.546	0.17
	Team	25	13.25	1.53	12.62	13.88			
UN-RA [s]	Individ.	31	17.06	1.64	16.46	17.67	4.62	0.036	0.59
	Team	24	16.02	1.96	15.19	16.85			
SC-CODS [s]	Individ.	29	16.81	1,29	16.32	17.31	5.18	0.027	0.62
	Team	26	15.96	1.48	15.36	16.56			
SC-RA [s]	Individ.	28	18.94	1.19	18.48	19.40	2.11	0.153	0.42
	Team	22	18.31	1.89	17.47	19.14			
LA-CODS [s]	Individ.	25	12.54	1.43	11.94	13.13	0.95	0.336	0.30
	Team	20	12.12	1.41	11.46	12.78			
LA-RA [s]	Individ.	25	14.41	1.12	13.95	14.87	7.72	0.008	0.86
	Team	19	13.37	1.37	12.71	14.03			

Frontal (FR), Universal (UN), Semi-circular (SC), Lateral (LA).

**Table 3 ijerph-17-00975-t003:** Linear sprint speed and lower limb explosive power and between-group comparisons.

Variables	Group	*N*	Mean	SD	95% Confidence Interval of the Mean		Mean Square	*F*	*p*	*d*
					Lower	Upper				
S15m [s]	Individ.	27	2.52	0.19	2.44	2.59	0.142	4.46	0.039	0.59
	Team	27	2.62	0.16	2.55	2.68				
F15m [s]	Individ.	27	2.07	0.18	1.99	2.14	0.114	3.58	0.064	0.53
	Team	27	1.98	0.17	1.91	2.04				
SLJ10m–right leg [s]	Individ.	27	2.69	0.39	2.54	2.83	0.789	5.82	0.020	0.78
	Team	26	2.42	0.31	2.29	2.54				
SLJ10m–left leg [s]	Individ.	29	2.65	0.39	2.50	2.81	0.971	7.76	0.007	0.67
	Team	26	2.41	0.34	2.27	2.55				

**Table 4 ijerph-17-00975-t004:** Within-group comparisons of change-of-direction speed (CODS) and reactive agility (RA) performance in different configurations.

Agility Performanc1			Paired Differences					*t*	df	*p*
			Mean	SD	SEM	95% Confidence Interval of the Difference				
						Lower	Upper			
Pair 1	FR-CODS–FR-RA [s]	Team	−2232.52	1571.86	282.32	−3109.08	−1955.95	−8.971	30	0.000
		Individ	–2547.68	1115.79	188.60	–2930.95	–2164.37	–13.51	34	
Pair 2	UN-CODS –UN-RA [s]	Team	−2929.00	2851.29	594.53	−4161.99	−1696.01	−4.927	22	0.000
		Individ	−4048.39	2447.37	439.56	−4946.09	−3150.69	−9.210	30	
Pair 3	SC-CODS –SC-RA [s]	Team	−2337.41	1444.59	307.99	−2977.90	−1696.92	−7.589	21	0.000
		Individ	−2171.50	1063.14	200.91	−2583.74	−1759.68	−10.81	27	
Pair 4	LA-CODS–LA-RA [s]	Team	−1314.89	652.67	149.73	−1629.47	−1000.32	−8.782	18	0.000
		Individ	−1873.56	1346.79	269.36	−2429.49	−1317.63	−6.96	24	

Frontal (FR), Universal (UN), Semi-circular (SC), Lateral (LA).

**Table 5 ijerph-17-00975-t005:** Within-group correlations of change-of-direction speed (CODS) and reactive agility (RA) performance in) different configurations.

Agility Performance			Mean	SD	*r*	*p*
**Pair 1**	FR-CODS	Team	16.16	1.77	0.805	0.000
		Individ.	16.68	1.75		
	FR-RA	Team	18.63	1.87	0.622	0.000
		Individ.	19.08	1.87		
**Pair 2**	UN-CODS	Team	13.25	1.53	−0.308	0.152
		Individ.	13.01	1.38		
	UN-RA	Team	16.02	1.96	−0.297	0.105
		Individ.	17.06	1.64		
**Pair 3**	SC-CODS	Team	15.96	1.48	0.666	0.010
		Individ.	16.81	1.29		
	SC-RA	Team	18.31	1.89	0.636	0.000
		Individ.	18.94	1.19		
**Pair 4**	LA-CODS	Team	12.12	1.41	0.890	0.010
		Individ.	12.54	1.43		
	LA-RA	Team	13.37	1.37	0.466	0.019
		Individ.	14.41	1.12		

Frontal (FR), Universal (UN), Semi-circular (SC), Lateral (LA).

**Table 6 ijerph-17-00975-t006:** Correlations between change-of-direction speed (CODS) and reactive agility (RA) in the frontal (FR), universal (UN), semi-circular (SC), and lateral (LA) configurations, sprinting and jumping performance, and anthropometric characteristics, * *p* < 0.05.

(**A**)																
**Team Athletes**								**Variables**	**Individual Athletes**							
**[8]**	**[7]**	**[6]**	**[5]**	**[4]**	**[3]**	**[2]**	**[1]**		**[1]**	**[2]**	**[3]**	**[4]**	**[5]**	**[6]**	**[7]**	**[8]**
			0.453 *					Body mass	–0.342 *			0.452 *			–0.405 *	
	–0.546 *				0.401 *			Body high						–0.432 *		
	–0.487 *				0.515 *			Leg length							–0.456 *	
	0.655 *				–0.449 *			SLJ0m (LL)				0.395 *				0.460 *
	0.566 *				–0.423 *			SLJ0m (RL)				0.410 *				
	0.626 *							S15m				0.465 *				0.506 *
	0.629 *				–0.491 *			F15m								
	[1] FR-CODS, [2] FR-RA, [3] UN-CODS, [4] UN-RA, [5] SC-CODS, [6] SC-RA, [7] LA-CODS, [8] LA-RA															
(**B**)																
**Team Athletes**								**Variables**	**Individual Athletes**							
**[8]**	**[7]**	**[6]**	**[5]**	**[4]**	**[3]**	**[2]**	**[1]**		**[1]**	**[2]**	**[3]**	**[4]**	**[5]**	**[6]**	**[7]**	**[8]**
			0.453 *				-	[1] FR-CODS	-			0.373 *				
						-	0.622 *	[2] FR-RA		-						
					-			[3] UNCODS			-					
				-				[4] UN-RA	0.373 *			-				0.483 *
			-				0.453 *	[5] SC-CODS					-			
0.624 *		-						[6] SC-RA						-		0.414 *
	-						0.606 *	[7] LA-CODS							-	0.466 *
-								[8] LA-RA				0.483 *		0.414 *	0.466 *	-

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
