# Peer review of "Evaluation of the Pre-Planned and Non-Planed Agility Performance: Comparison between Individual and Team Sports"

_ijerph, 2020, doi:10.3390/ijerph17030975_

Round 1
Reviewer 1 Report
1. The introduction is incomplete. Please go deeper.
2. Please, include Cohen’s d in every analysis.
3. The discussion is incomplete. You must make an effort to discuss as a specialists.
4. Further study applications are necessary.
5. Errors are all over the paper. It must be corrected, for instance:
‘Indyvidual’
‘reactive agility ant’
Reviewer 2 Report
General comments
The article entitled “Evaluation of the pre-planned and non-planed agility performance: comparison between individual and team sports” aims to compare the agility performance between groups of individual and team sports participants by assessing the ability to change of direction and the reactive agility. The purpose is interesting and relevant to the area. However, there are serious methodological flaws regarding the sampling procedures. In general, this looks like a case-study instead of an experimental one since the groups are too heterogeneous and there are not sample size estimation procedures. If a clear rationale for the sampling procedures is presented, making sure that it is possible to generalize the current results, I think this article has the potential to be accepted. On the other hand, without this rationale, I recommend its rejection.
Specific comments
Introduction.
Introduction, in general, is regular. The research problem is not well addressed. Since individual and team sports’ athletes are in different contexts (they practice different sports), what is the rationale behind comparing them? What contribution is expected from it? It seems quite obvious and non-relevant to address that practitioners from different modalities are, in fact different.
Without support from previous literature analyzing agility performance from athletes, it is impossible to establish a hypothesis. The current hypothesis was proposed without any support from the literature. This is a mandatory issue to be addressed by the authors.
“The latest approaches to Teaching Games for Understanding (TGfU) support a thematic approach to teaching games. Instead of teaching and concentrating on one game”. Since you are talking about some previous TGFu approaches, it is mandatory to reference it.
Agility has been defined without the consideration of the cognitive component (third paragraph). Later in the paragraph, this definition was contested. If the authors consider that the current definition must take into account the cognitive component, presenting the other definition is no longer necessary. I recommend you to go straight to the current and valid definition, avoiding confusing the readers.
“In sports situations, RA is more common than preplanned CODS since sport-specific situations are frequently”. This is true just for open sports (such as team sports). Individual and closed sports do not have this characteristic (like track and field).
Methods
How the sample size was estimated? Is the current sample enough to generalize the results?
Your sample is too heterogeneous. For this reason, generalizations are not plausible and the results may not represent the real phenomena. How do you assure that the current results are due to the group factor (individual versus team sports) instead of the characteristics of these specific groups? Without a clear justification for this, the article must be rejected.
What is the criterion for the 4-years cut regarding the previous experience? This is not a usual value.
How did you assure that the participants were familiar with the testing procedures (how did you reduce the familiarization bias)? Did you calculate the SEM and checked if the performance was stable?
Please present the thresholds for interpreting the effect sizes.
Please inform which ICC has been used (see Weir, 2005
Results
Table 1: the column “mean” is empty.
The root squares are not necessary.
Notes are missing in all the tables since abbreviations were used for all variables.
The correlation table has poor visual quality. Please, increase it.
Discussion
Individual athletes performed better or these specific individual athletes performed better? The same question pointed out previously…
The third paragraph is just speculation and must be avoided. You only split participants based on the individual/team status of the sport, so you cannot consider specific characteristics of sports to justify the results. At this point, for example, you used specific characteristics of tennis to explain the results, and these characteristics are obviously not present in track and field competitions, but you have athletes from these modalities within the same group.
Conclusion
The first sentence is not supported by the results since the authors did not assure that the data is generalizable. You cannot point out that all individual athletes will always perform better than team sports athletes since many details when sampling the participants were neglected.
References
The reference 6 is not rightly presented.
Round 2
Reviewer 2 Report
Dear authors,
I would like to congratulate you on the time spent to solve some issues regarding the article. However, in my opinion, the most relevant issues were not addressed, so I am in favor to reject the current submission. The non-addressed relevant points are:
The absence of a clear definition for splitting groups. Within the participants of the individual sport, you have players from open games (like tennis and judo) and closed games (like track and field). It seems obvious that these players have completely different demands regarding agility since individual closed sports do not require changing the direction after an external stimulus while individual open sports require. So, you have two groups inside the same, which is a serious methodological flaw. The sample was not statistically estimated, and this question was not even answered in the cover letter. You did not argue about the potential for generalizing the results considering the high specificity of the sample (again: is it a case study?) The 4-year period for selecting subjects has no external validity (it only refers to the specific context from which the sample was extracted - again, generalization is not possible).Author Response
Please see the attachment

Round 3
Reviewer 2 Report
Approved in its current form.